# DNA Methylation of Genes Involved in the HPA Axis in Presence of Suicide Behavior: A Systematic Review

**DOI:** 10.3390/brainsci13040584

**Published:** 2023-03-30

**Authors:** Diana María Dionisio-García, Alma Delia Genis-Mendoza, Thelma Beatriz González-Castro, Carlos Alfonso Tovilla-Zárate, Isela Esther Juarez-Rojop, María Lilia López-Narváez, Yazmín Hernández-Díaz, Humberto Nicolini, Viridiana Olvera-Hernández

**Affiliations:** 1División Académica de Ciencias de la Salud, Universidad Juárez Autónoma de Tabasco, Jalpa de Méndez 86205, Mexico; 2Hospital Psiquiátrico Infantil “Juan N. Navarro”, Tlalpan 14080, Mexico; 3Laboratorio de Genómica de Enfermedades Psiquiátricas y Neurodegenerativas, Instituto Nacional de Medicina Genómica, Ciudad de México 14610, Mexico; 4División Académica Multidisciplinaria de Jalpa de Méndez, Universidad Juárez Autónoma de Tabasco, Jalpa de Méndez 86205, Mexico; 5División Académica Multidisciplinaria de Comalcalco, Universidad Juárez Autónoma de Tabasco, Comalcalco 86650, Mexico; 6Hospital Chiapas nos une “Jesús Gilberto Gómez Maza”, Secretaría de Salud, Tuxtla Gutiérrez 29045, Mexico

**Keywords:** DNA methylation, HPA axis, epigenetic, suicide behavior, stress

## Abstract

DNA methylation in genes of the hypothalamic–pituitary–adrenal (HPA) axis has been associated with suicide behavior. Through a systematic review, we aimed to evaluate DNA methylation levels of the genes involved in the HPA pathway and their association with suicide behavior. A search of articles was performed using PubMed and Science Direct, EBSCO. The terms included were “DNA methylation”, “suicide”, “epigenetics”, “HPA axis” and “suicide behavior”. This systematic review was performed by the Preferred Reporting Items for Systematic Review and Meta-Analyses (PRISMA) statement. Six studies comprising 743 cases and 761 controls were included in this systematic review. The studies included individuals with suicide ideation, suicide attempts or completed suicide and childhood trauma, post-traumatic stress disorder (PTSD), or depression. One study reported hypermethylation in GR in childhood trauma, while two studies found hypermethylation of NR3C1 in childhood trauma and major depressive disorder (MDD). Only one study reported hypermethylation in BNDF in people with MDD. FKBP5 was found to be hypermethylated in people with MDD. Another study reported hypermethylation in CRHBP. SKA2 was reported to be hypermethylated in one study and another study found hypomethylated both in populations with PTSD. CRHR1 was found to be hypermethylated in people with MDD, and the last study found hypomethylation in CRH. Our result showed that patients with suicidal behavior showed a DNA methylation state of genes of the HPA axis in association with psychiatric comorbidity and with adverse events. Genes of the HPA axis could play a role in suicidal behavior associated with adverse events and pathologies. As a result, DNA methylation levels, proteins, and genes involved in the HPA axis could be considered for the search for biomarkers for the prevention of suicidal behavior in future studies.

## 1. Introduction

Suicide is a public health problem around the world. According to the World Health Organization (WHO) every year, more than 700,000 individuals die by suicide worldwide [1]. In the United States, 1.20 million suicide attempts were reported in 2020 [2]. The factors involved in suicide development are diverse and complex. Until today, the risk of suicide is associated with clinical, psychological, environmental, and genetic factors. Some studies have found an association between suicidal behavior and alterations in the Hypothalamic-Pituitary–Adrenal axis (HPA), a system that is related to stress and suicidal behavior [3].

The literature has found some biological factors associated with suicidal behavior, such as the low function of the serotonergic systems that was present in people who attempted suicide [1], as well as the dopamine system, which is associated with suicide attempt risks in some individuals [2,3]. Studies also demonstrate that chronic stress has been associated with alterations in the HPA axis and suicidal behavior [4,5,6,7]. Several studies indicated hypercortisolemia in patients with suicidal behavior [8,9,10], while others reported lower levels of cortisol in response to stress tests [6,11,12,13]. No conclusive association has been established between cortisol levels and suicidal behavior, although some studies are searching for epigenetic changes in people with suicidal behavior. The evidence reported that environmental factors causing health problems and epigenetic mechanisms could be associated with the HPA axis and suicidal behavior [14].

Epigenetics is the study of modifications that affect gene expression but without changes in DNA sequences [15]. One form of epigenetic change is DNA methylation, which consists of the addition of methyl groups to DNA segments, which leads to reduced gene expression in these DNA regions [16] and prevents the binding of transcriptional activators [17]. DNA methylation sites have been associated with suicide behavior. As an example, people who had a suicide attempt showed lower cortisol concentrations and lower GR or *NR3C1* (α isoform) mRNA [8], which would be linked to *NR3C1* methylation. Moreover, hypomethylation in the *FKBP5* gene has been found in veterans exposed to combat trauma who present post-traumatic stress disorder (PTSD) [18]. This was also observed in people exposed to the holocaust as well as in their offspring [19].

Likewise, another study conducted on patients with major depressive disorder (MDD) and suicidal ideation presented hypomethylation of the *FKBP5* gene when compared to patients with MDD and no suicidal ideation [20]. Major depressive disorder is an important disabling mental disorder, and some risk factors that are associated with the development of MDD [4] are a family history of MDD [5], deficits in how individuals process reward [6], substance use disorder [7], sexual abuse [8], social isolation and others [9].

In the case of depression, some epigenetic mechanisms could mediate the risk to have depression, not only DNA methylation or microRNAs (miRNAs) and histone modifications. [10,11]. For example, studies showed altered levels of specific miRNA in the peripheral tissue of patients with MDD [12], and in brain tissue another study found miR-1202 downregulated in the PFC of MDD patients who died by suicide [13]. As for histone modifications, another study found increased global acetylation of histone 3 at lysine 14 (H3K14ac), due to the above in the nucleus accumbens of patients with MDD reported downregulation of HDAC2 [14]. 

Unfortunately, until now there are no conclusive results. Therefore, the roles of the genes that are involved in the HPA axis and suicidal behavior need to be elucidated.

### Aims to Clarify

Additionally, we also aimed to determine if the DNA methylation levels of genes that participate in HPA axis activation could be associated with an increased risk of suicide behavior.

## 2. Materials and Methods

### 2.1. Search Strategy

A search of relevant studies that investigated the association between DNA methylation levels of genes involved in HPA axis activation and suicidal behavior was performed. The search was conducted up to April 2022 using a predetermined protocol that follows the *Preferred Reporting Items for Systematic Review and Meta-Analyses* (PRISMA) statement. This study was registered with the International Prospective Register of Systematic Reviews (PROSPERO) (registration number 348748).

The systematic search included articles that published between 2009 and 2018. This search was performed using three electronic databases (PubMed, Science Direct, and EBSCO). Two authors (DMD-G and TBG-C) independently identified records and extracted data; in case of disagreement, a third researcher intervened and decided. Several combinate questions of terms were used for the search: “methylation and suicide”, “methylation and stress”, “epigenetics and suicide”, “epigenetic and HPA axis”, “methylation and suicide behavior”, “epigenetic and suicide behavior”, “methylation and cortisol”, “epigenetic and cortisol”. These words were combined in different ways to retrieve the summaries. All the genes that were included had to codify a protein that participated in HPA axis activation. (Figure 1).

### 2.2. Inclusion Criteria and Data Extraction

To select suitable studies, the inclusion criteria were (1) original article, (2) association studies of suicide and methylation, (3) studies published in English, and (4) studies published in peer reviewed journals. From each included article, we retrieved authors, year of publication, country of origin, type of suicide behavior, study design, method of gene expression analysis, methylation measuring method, sample, diagnostics of cases and controls, number of cases and controls, sociodemographic characteristics, genes studied, and main findings.

### 2.3. Exclusion Criteria

The exclusion criteria were duplicated publications, studies of cases only or case reports, and papers that did not have enough data available.

### 2.4. Quality Assessment of Primary Studies

The quality of the studies was evaluated by the Newcastle–Ottawa scale (NOS). This international scale includes three criteria: (1) Selection of the studies, (2) comparability, and (3) outcomes. The scores range from 0 to 9; scores 7–9 for high quality, scores 4–6 for moderate quality and 0–3 for poor quality.

### 2.5. Data synthesis

The information extracted from the studies was organized in tables. The phenotypes included in this systematic review were suicide ideation, suicide attempts, and death by suicide.

## 3. Results

### 3.1. Characteristics of Eligible Studies

A total of 15 articles of interest were reviewed for title and abstract; of those, 9 articles were excluded, because they did not evaluate suicide behavior. Finally, we included six studies in this systematic review (Figure 2). Two studies used brain tissue to measure DNA methylation levels in individuals who died by suicide [21,22] and four studies used blood to measure DNA methylation in individuals with suicidal ideation and suicide attempts [20,23,24,25].

### 3.2. Characteristics of the Studies

The studies included in this systematic review were published between 2009 and 2018. The sample included cases of individuals with suicidal behavior and healthy controls. The number of cases in each study ranged from 14 to 706 individuals, and the number of controls ranged from 20 to 759. All the studies evaluated males and females. Most of the studies were conducted in United States of America (USA) (*n* = 3, 50%) [20,23,25], followed by Canada (*n* = 2, 33.33%) [21,22] and Sweden (*n* = 1, 16.6%) [24]. One study used individuals from the Grady Trauma Project and Johns Hopkins Center for Prevention Research Study [25]. Table 1 shows the characteristics of the studies included. According to the NOS score, the quality of the studies ranged between 7 and 9. See Table 1.

Two studies [21,22] explored the methylation levels using cerebral tissue [21] and the hippocampus and Broadman area 24 [22]. Two studies used peripheral blood [20,23], one used whole blood [24] and one used whole blood and saliva [25]. The CPG site with changes in epigenetics was *SKA2* CpG 13989295 [23,25] *GR* Glucocorticoid receptor CpG6, CpG8, CpG11 [21], *N3CR1* CpG30, CpG32 [22], *CRH* Corticotropin-Releasing-Hormone CpG19035496 and CpG23409074 [24]. See Table 2.

A summary and comparative table of the methylation state of genes included in the present study of the influence in the activation of the HPA axis and reported in childhood trauma, post-traumatic stress disorder (PTSD), depression, and psychiatric risk are shown in the Table 3.

Labonte B. et al., 2012 [21] analyzed the first eight coding exons of human glucocorticoid receptor (HGR). The exon named *HGRB1* (CPG6 site) showed hypermethylation in individuals who were abuse victims and died by suicide when compared to healthy individuals for (*p* < 0.005) and CPG8 site (*p* < 0.05). By contrast, hypomethylation was observed for CpG11 site (*p* < 0.005). Patients with childhood trauma and death by suicide showed hypermethylation in the CpG8, CpG9 site when compared to those without childhood trauma (*p* < 0.005) and in CpG12 site of the *HGRC1* exon. By contrast, the CpG13 site was significantly hypomethylated. The findings in terms of the HGR1H exon were hypomethylation in individuals in abuse victims dead by suicide group compared to the non-abused dead by suicide and control groups for the CpG2 site, CpG5 site and CpG10 site (non-abused dead by suicide: *p* < 0.01, healthy controls: *p* < 0.05). With regard to the CpG1 site, a significant lower methylation was observed in the abused dead by suicide group (*p* < 0.05) compared to the controls without any significant difference between the abused dead by suicide and non-abused dead by suicide groups. Finally, a lower methylation at the CpG3 (*p* < 0.005), CpG7 (*p* < 0.05), CpG8 (*p* < 0.05), and CpG12 sites (*p* < 0.01) was observed in the abused dead by suicide group compared to the non-abused dead by suicide group, whereas no significant difference was found between the abused dead by suicide group and the healthy controls.

Other study conducted by McGowan [22] used the same overlapping cohort with Labonte, with a similar clinical group from the same bank brain in Canada. This compared the methylation of the *NR3C1* gene in the postmortem hippocampus obtained from three groups: first, individuals who died by suicide and had a history of childhood trauma; second, individuals who died by suicide without a history of childhood trauma; and third, the individuals who died by any other cause. The analysis of methylation of the F exon showed no statistical differences between groups. Nonetheless, the analysis of the 1F exon showed hypermethylation in individuals who died by suicide and had history of childhood trauma when compared to non-abused individuals who died by suicide.

One study analyzed the methylation of the *SKA2* gene in patients with post-traumatic stress disorder [25]. They found a significant interaction between the scores of the Child Trauma Questionnaire (CTQ) and *SKA2* hypermethylation, after adjusting age, sex, race, and lifetime history of substance abuse (β-value 0.100, *p* = 0.037). Furthermore, *SKA2* DNA methylation interacted with the scores of CTQ to predict a lifetime suicide attempt of 0.76 and 0.73 (95% confidence interval (CI): 0.6–0.92, *p* = 0.003, and CI: 0.65–0.78, *p* = 0.0001). The interaction between trauma and methylation of *SKA2* DNA was stronger when trauma resulted from emotional abuse than from physical or sexual abuse (β-value 0.0036, *p* = 0.01).

A study in veterans analyzed the methylation levels in *SKA2* DNA [23]. They used the PTSD scale to assess the frequency and severity of the 17 symptoms of PTSD reported in the DSM-IV. The regression analysis demonstrated a significant association between the *SKA2* hypomethylation and current suicidal thoughts and behaviors (β = 0.27, *p* = 0.014); however, the *SKA2* methylation was not significantly associated with current or lifetime PTSD severity scores (|βs| < 0.05, *p* ≥ 0.66).

Roy B. et al., 2017 found that *BNDF, FKBP5, CRHBP,* and *NR3C1* gene promoters were hypermethylated in individuals with major depression disorder with and without suicidal ideation compared to the controls [20]. The results showed an enrichment of CpG islands in the promoter regions of stress related genes and were significantly different between MDD-suicidal ideation and the healthy controls: *BNDF* (*p* = 0.039), *FKBP5* (*p* = 0.035), *CRHBP* (*p* = 0.039) and *NR3C1* (*p* = 0.025), but not *CRHR1* (*p* = 0.37); between MDD-non-suicidal ideation and the healthy controls, significant differences were noted in: *BNDF* (*p* = 0.31), *FKBP5* (*p* = 0.13), *CRHBP* (*p* = 0.72), *CRHR1* (*p* = 0.96), and *NR3C1* (*p* = 0.15), and this is associated with the decreased expression of BNDF, FKBPF5 (variants 1,2 y 3) NR3C1 genes in these patients.

Finally, Jokinen et al. [24] stratified the sample into high-risk and low-risk groups based on the severity of the suicide attempt and identified the hypothalamic–pituitary– adrenal axis coupled CpG-sites showing hypomethylation shifts linked to the severity of the suicide attempt. Two *CRH*-associated CpG sites were significantly hypomethylated in patients with severe suicide attempts cg19035496 (*p* = 0.049) and cg23409074 (*p* = 0.016).

## 4. Discussion

Our systematic review analyzed the DNA methylation in genes related to the HPA axis in suicide attempts, suicidal ideation, and deaths by suicide in association with childhood trauma, PTSD, major depressive disorder, and psychiatric risk. The results of these articles demonstrated epigenetic changes in genes related to the HPA axis and were associated with suicide behavior. Suicide is a worldwide health problem. Globally, it is the fourth leading cause of death in people aged 15–29 years [2]. The factors associated with the development of suicidal behavior are diverse and complex; in the last years, the HPA axis alterations have been highly related to suicide [5,26,27]. This axis responds to situations that generate stress and alter homeostasis [28].

In this review, we found that some genes that are part of the HPA axis activation showed epigenetic changes in people with suicidal behavior [29,30]. In the case of the *CRH* gene, it expresses the hormone that helps establish the initial response of the HPA axis to a stressful event [31]. This corticotropin-releasing hormone receptor has been observed hypomethylated and its expression increased in individuals who attempted suicide [24]. Moreover, elevated *CRH* levels have been found in extra hypothalamic brain regions of individuals with depression who died by suicide [32]. In the case of the *FKBP5* gene, which encodes the FKBP51 protein (cis-trans prolyl isomerase), its most important function is to influence binding affinity in glucocorticoid receptor signaling [33], which is hypomethylated in people who attempted suicide and also in patients with depression and suicidal ideation [20,24]. Joined studies found that alterations in methylated levels of *CRH* were associated with post-traumatic stress disorder, bipolar disorder, suicide attempts, and major depressive disorder in HIV patients [34]. On the other hand, the *NR3C1* expresses the cortisol receptor and mediates glucocorticoid actions, and it also functions as a hormone-dependent transcription factor that regulates the expression of glucocorticoid-responsive genes [35], which is hypermethylated in people with suicide ideation and depression [20], as well as the *CRHBP* gene, which codifies the corticotrophin-releasing factor-binding protein, regulates HPA axis function and stimulates synthesis and secretion of proopiomelanocortin derived peptides [36]. This gene was found to be hypermethylated in patients who attempted suicide and patients with major depressive disorder with and without suicidal ideation [20,24]. Last, the *SKA2* gene encodes a protein involved in metaphase plate maintenance and spindle checkpoint silencing during mitosis [37] and has been implicated in enabling glucocorticoid receptor nuclear transactivation [38]. A decreased expression of the *SKA2* gene has been observed in the prefrontal cortex of people who died by suicide [39]. In addition, hypermethylation of the *SKA2* gene has been observed in patients with post-traumatic stress disorder and suicidal behavior [23,25].

The findings of this systematic review suggest that genes of the HPA stress response pathway are significantly associated with suicidal behavior [40]. As the DNA methylation of these genes is altered, the genes may be influencing a dysregulation of HPA axis activity as measured by altered cortisol levels [6,41]. Interestingly, some GWAS studies have reported a significant association between HPA axis genes such as *CRHR1, CRHR2, FKBP5, CRHBP, NR3C1, AVPR1B,* and suicidal behavior [42,43]. Overall, this evidence suggests a role of the HPA axis genes and suicide behavior. Whether the HPA axis response is moderated or altered, ultimately it influences the suppression of cortisol in its response to stress. Therefore, for this pathway we also need to search and identify probable biological markers that will be capable of identifying the risk of suicide in our population, serving as prevention for the population.

We recognize that there are previous reviews that identified the alterations in epigenetics with the HPA axis associated with suicidality, highlighting the importance of this axis’s findings and suicide [44] or genetic modifications related to defective regulation of the HPA axis in suicidal behavior [45,46]. However, ours is the first study to aggregate reports of genes involved in the HPA pathway.

The most outstanding findings were in suicide completers with childhood trauma, *NR3C1* gene reported hypermethylation, and also in the case of suicidal ideation with MDD, *NR3C1, BNDF, FKBP5, CRHBP,* and *CRHR1* found in hypermethylation. Likewise, *SKA2* genes studied in PTSD and suicide attempters were hypermethylated. Furthermore, in suicide attempters and psychiatric risk the reported CRH gene was hypomethylated. At the same, in patients with suicidal thoughts and PTSD, the *SKA2* was found to be hypomethylated. Based on these findings, it might be suggested that DNA methylation depends on suicidal behavior and pathology.

Our study has some limitations. First, the studies included broad diverse pathologies: childhood trauma, post-traumatic stress disorder, and depression. Perhaps, it could be a limitation, but pathologies that are due to childhood events of traumas are associated with epigenetic changes [47]. Therefore, the studies used different inclusion and exclusion criteria for their participants. Second, each study evaluated different behaviors: death by suicide, suicide behavior, suicide ideation, and suicide attempts. Finally, some studies measured DNA methylation levels in differences tissues.

In conclusion, in this systematic review we found that the DNA methylation state of some genes essential for the activation of the HPA axis is altered. This state of DNA methylation is related to suicide behavior, suicide attempts, or suicide risk in patients with different psychiatric comorbidities and with adverse events. As a result, the HPA axis could be considered as a possible field to search biological markers that help prevent suicide behavior. Because genes that have changes in their methylation levels are involved in the activation of the HPA axis and in the creation of the hormone cortisol, they could share this common pathway.

## Figures and Tables

**Figure 1 brainsci-13-00584-f001:**
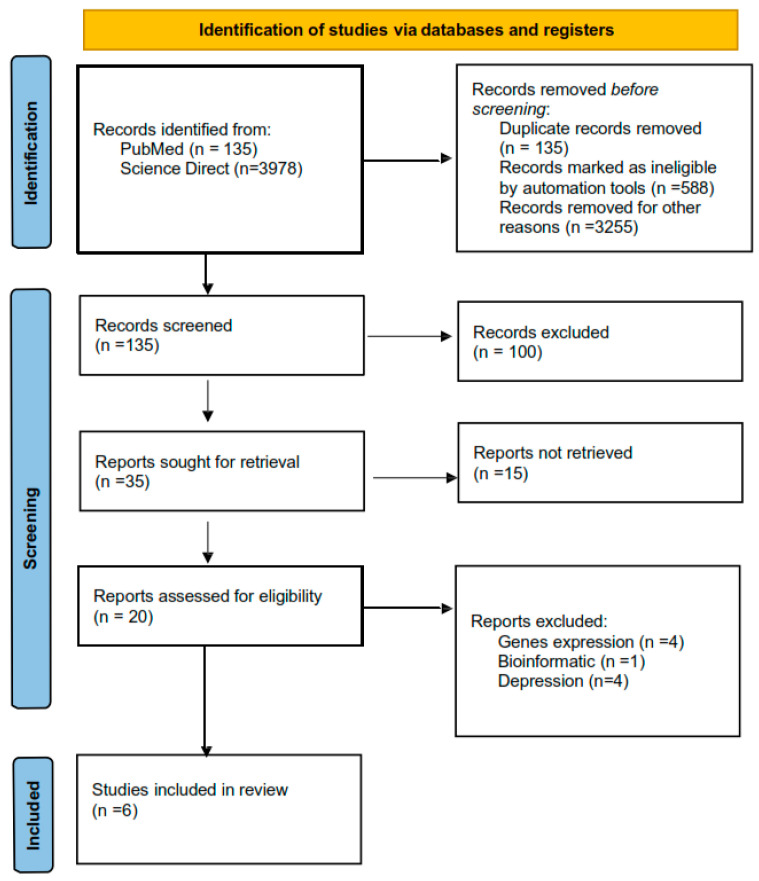
PRISMA flow chart that includes all the processes of choosing articles.

**Figure 2 brainsci-13-00584-f002:**
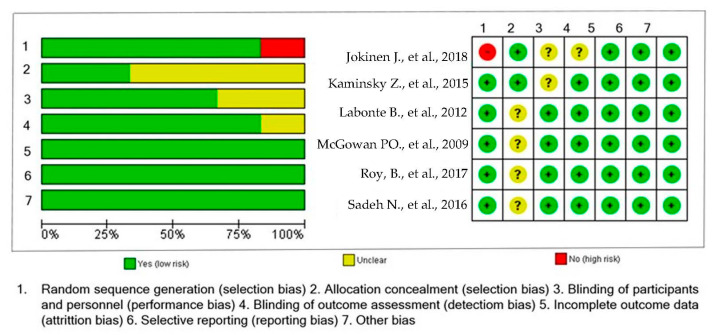
Risk of bias summary: a review of authors’ judgments about risk of bias item for each included study [20,21,22,23,24,25].

**Table 1 brainsci-13-00584-t001:** Characteristics of the included publications that evaluated methylation of HPA genes expression (mRNA) in the pathogenesis of suicide behavior.

Brain Samples								
Author and Year	Origin	Genes	Diagnostics	Cases		Controls		NOS
			Cases/ Controls	Median Age	Male/ Female	Median Age	Male/ Female	
Labonte B. 2012 [21]	French- Canadian Caucasian	GR	ELS/ non-psychiatric	39.05 ± 2.4	42/0	39.8 ± 3.9	14/ 0	9
McGowan PO, 2009 [22]	French-Canadian	N3CR1	Childhood trauma/ non-psychiatric	34.2 ± 10/33.8 ± 11	24/0	35.8 ± 12	12/ 0	9
Blood samples								
Sadeh N. 2016 [23]	White, non-Hispanic	SKA2	PTSD	52.4 ± 10.7	302/104			9
Roy B, 2017 [20]	Caucasian/African American	BNDF NR3C1 FKBP5 CRHBP CRHR1	MDD with suicidal ideation/ Healthy controls MDD without suicidal ideation	42.90 ± 2.99	6/8	-	6/ 14	9
Kaminsky Z, 2015 [25]	African American	SKA2	PTSD with current suicidal ideation PTSD with lifetime suicide attempts/ Healthy controls	52.4	- -	-	- -	7
Jokinen J. 2018 [24]		CRH CRHBP CRHR1 FKBP5 NR3C1	Suicide attempts/ Healthy controls	35.16 ± 12.3	16/15	33.6 ± 12.2	12/ 45	8

GR: Glucocorticoid Receptor; N3CR1: SKA2: Spindle and Kinetochore Associated Complex Subunit 2; BNDF: Brain-Derived Neurotrophic Factor; FKBP5: FKBP Prolyl Isomerase 5; CRHBP: Corticotropin-Releasing Hormone Binding Protein; CRHR1: Corticotropin-Releasing Hormone Receptor 1; CRH: Corticotropin-Releasing Hormone; CRHBP: Corticotropin-Releasing Hormone Binding Protein; ELS: Early-Life Stress; MDD: Major Depressive Disorder.

**Table 2 brainsci-13-00584-t002:** Characteristics of the studies included in the systematic review.

Author and Year	Country	Population Data (N)	Main Outcomes
		Cases/Controls	
Labonte B, 2012 [21]	Canada	42/14	Hypermethylation of the GR gene in the hippocampus CPG6 (SA: *p* < 0.05; SNA: *p* < 0.005), CPG8 (SA: *p* < 0.05, SNA: *p* < 0.05), CPG11 (SA: *p* < 0.005; SNA: *p* < 0.005).
McGowan PO, 2009 [22]	Canada	24/12	RNAm transcripts bearing the glucocorticoid receptor 1 slice variant and increased cytosine Methylation NR3C1 promotes.
Roy B, 2017 [20]	United States of America	14/20	Hypermethylation in MDD patients with and without suicidal ideation gene promoters BNDF, NR3C1, FKBP5, CRHBP. CPG30 (*p* < 0.001), CPG32 (*p* < 0.001) deaths by suicide with no history of childhood trauma and controls.
Kaminsky Z, 2015 [25]	United States of America	67 CSI /99 LSAC/ 337/ 321	SKA2 methylation levels also predicted higher rates of current suicidal thoughts and behaviors.
Jokinen J, 2018 [24]	Sweden	31/ 57	Hypomethylation Two CpG CRH cg19035496 and cg23409074, in suicide attempters and hypermethylated cg19035496 high general psychiatric risk score.
Sadeh N, 2016 [23]	United States of America	466	Methylation at the CpG locus cg13989295 was associated with higher rates of suicidal thoughts and behaviors.

CSI: current suicidal ideators; LSAC: lifetime suicide attempt cases, #CpG: site when cytosine and guanine appear consecutively on the same strand of nucleic acid. SA: Suicide Attempted. MDD: Major Depressive Disorder.

**Table 3 brainsci-13-00584-t003:** The methylation state of genes that influence the activation of the HPA axis and reported in ELS, PTSD, MDD, and psychiatric risk.

Gene	Childhood Trauma			Post-Traumatic Stress Disorder			Major Depressive Disorder			General Psychiatric Risk			References
	N	Hypo	Hyper	N	Hypo	Hyper	N	Hypo	Hyper	N	Hypo	Hyper	
GR	1	0	1	0	0	0	0	0	0	0	0	0	Labonte B, 2012 [21]
NR3C1	1	0	1	0	0	0	1	0	1	1	0	0	McGowan PO, 2009 [22] Roy B, 2017 [20] Jokinen J. 2018 [24]
BNDF	0	0	0	0	0	0	1	0	1	0	0	0	Roy B, 2017 [20]
FKBP5	0	0	0	0	0	0	1	0	1	1	0	0	Roy B, 2017 [20] Jokinen J. 2018 [24]
CRHBP	0	0	0	0	0	0	1	0	1	1	0	0	Roy B, 2017 [20] Jokinen J. 2018 [24]
SKA2	0	0	0	2	1	1	0	0	0	0	0	0	Kaminsky Z. 2015 [25] Sadeh N. 2016 [23]
CRHR1	0	0	0	0	0	0	1	0	1	1	0	0	Roy B. 2017 [20] Jokinen J. 2018 [24]
CRHR2	0	0	0	0	0	0	0	0	0	1	0	0	Jokinen J. 2018 [24]
CRH	0	0	0	0	0	0	0	0	0	1	1	0	Jokinen J. 2018 [24]

Hypo: hypomethylation; Hyper: hypermethylation.

## Data Availability

Not applicable.

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
