# Peer review of "DNA Methylation of Genes Involved in the HPA Axis in Presence of Suicide Behavior: A Systematic Review"

_brainsci, 2023, doi:10.3390/brainsci13040584_

Round 1

Reviewer 1 Report

The manuscript entitled "DNA Methylation of genes involved in HPA axis in presence of suicide behavior: a systematic review" claims to present evidence that DNA methylation in genes of the HPA axis has been associated with suicidal behavior. Even though the purpose of the submitted manuscript is interesting, the Reviewer found some significant shortcomings that must be improved (major revision).

1.       Abstract – has to be improved – too much of abbreviations, lack of

2.       Introduction. The first sentence seems to be odd "Childhood maltreatment is physical and emotional mistreatment"

3.       The whole last paragraph has to be changed.

4.       Figure 2 – has to be improved

5.       Figure 3 – it doesn't fit here. It has to be an illustration of the text. In present form does not match the text, or the text does not match the figure.

6.       It looks like a graphical abstract.

7.       Tables 1 and 2 – they don't properly show the data, is tough to read and analyze them

8.       A list of shortcuts is required. All abbreviations should have an expansion, but they don't. 

Author Response

REVIEWER 1

The manuscript entitled "DNA Methylation of genes involved in HPA axis in presence of suicide behavior: a systematic review" claims to present evidence that DNA methylation in genes of the HPA axis has been associated with suicidal behavior. Even though the purpose of the submitted manuscript is interesting, the Reviewer found some significant shortcomings that must be improved (major revision).

  1. Abstract – has to be improved – too much of abbreviations, lack of

Response: Thank you for your comment.  Now, we added all the significant of the abbreviatios, we used in this manuscript.

Change in the manuscript.

Page 1, Line 21. Hipothalamic-Pituitary-Adrenal (HPA)

Page 1, Line 26. Preferred Reporting Items for Systematic reviews and Meta-Analyses (PRISMA)

Page 1, Line 29. Post traumatic-stress disorder (PTSD)

  1. Introduction. The first sentence seems to be odd "Childhood maltreatment is physical and emotional mistreatment"

Response: Thank you, we changed the first sentence

Change in the manuscript

Page 2, line 72-74. In another hand, childhood trauma consist in physical and/or emotional mistreatment, sexual abuse, neglect, and negligent treatment in childhood, as well as their commercial or other exploitation. 

  1. The whole last paragraph has to be changed.

Response: We thankful your comment. Now, we had rewritten the last paragraph.

Change in the manuscript.

Page 2, line 75-81

Childhood trauma has been associated with the risk of mental diseases including suicidal behavior (Afifi et al 2016).

As evidence suggests that childhood adversity generates epigenetic modifications like alterations in behavior, presenting psychiatric disorders in adulthood [14], it is necessary to investigate more about this topic to develop strategies to prevent suicide behavior . The aim of this systematic review was to evaluate if the DNA methylation in genes of the HPA axis is associated with suicide behavior.

  1. Figure 2 – has to be improved

Response: Thank you for your comment, we modified the figure.

  1. Figure 3 – it doesn't fit here. It has to be an illustration of the text. In present form does not match the text, or the text does not match the figure.

Response: Thank you so much for your suggestion, we think it would be a good idea to delete the figure.

  1. It looks like a graphical abstract.

Response: Thank you so much for your suggestion, we think it would be a good idea to delete the figure in the text and we can use for the graphical abstract.

  1. Tables 1 and 2 – they don't properly show the data, is tough to read and analyze them

Response: Thank you so much for the comment, we restructured the tables that we used and try to clarify all the data.

Page 6 and 7.

  1. A list of shortcuts is required. All abbreviations should have an expansion, but they don't. 

Response: We apologize for this mistake; we added all the significance of the abbreviations.

Reviewer 2 Report

This is a very interesting systematic review focused on the association of DNA methylation of genes involved in HPA axis. The paper is well-written and of interest for the readers; however, several changes are recommended.

ABSTRACT

1- The methods of the paper should be explained after the description of the main aims. It should be stated that the authors conducted a Systematic Review.

2-Six studies were included: how many of them included the methylation of each genes reported in the results?

3- The conclusion of the abstract is correct. It adds information for further studies.

INTRODUCTION

1- Other biological factors associated with suicide behavior should be presented in the introduction. I recommend to expand the first part concerning the association between hormones, methylation and suicide.

2-The main objective of the paper should be better clarified. I recommend to add a subsection called 1.1. Aims to clarify them.

MATERIAL AND METHODS

1- In section 2.2 the authors report the inclusion criteria and variables recorded. What about exclusion criteria?

RESULTS

1- Section 3.3. should not be presented as a separate section with figures and tables. They should be presented in the body of the manuscript where appropiate. 

2- PRISMA flowchart should be presented in the methods section. This figure should be earlier presented.

CONCLUSIONS

1- A separate section for the conclusions is needed. Conclusions should not be referencing a Figure.

Author Response

REVIEWER 2

This is a very interesting systematic review focused on the association of DNA methylation of genes involved in HPA axis. The paper is well-written and of interest for the readers; however, several changes are recommended.

ABSTRACT

1- The methods of the paper should be explained after the description of the main aims. It should be stated that the authors conducted a Systematic Review.

Response: Thank you so much for your comments. We added the number registration in PROSPERO and completed PRISMA checklist.

Change in the manuscript:

Page 2, line 82.

This study was register with the International Prospective Register of Systematic Reviews (PROSPERO) (registration number 348748.

2-Six studies were included: how many of them included the methylation of each genes reported in the results?

Response: Thank you so much for your comments. We explained better what number of studies found genes hypomethylated or hypermethylated.

Changes in the manuscript

Page 1, line 30-36

One study reported hypermethylation in GR in childhood trauma, two studies found hypermethylation of NR3C1 in childhood trauma and Major depressive disorder (MDD) Only one study reported hypermethylation in BNDF in people with MDD. FKBP5 was found hypermethylated in people with MDD. Another study reported hypermethylation in CRHBP. SKA2 was reported hypermethylated in one study and another study found hypomethylated both in populations with PTSD. CRHR1 was found hypermethylated in people with MDD. And the last study found hypomethylation in CRH.

3- The conclusion of the abstract is correct. It adds information for further studies.

Response: Thank you so much for your comments

Changes in the manuscript

Page 1, line 39-41

Then DNA methylation levels, proteins, and genes involved in the HPA axis could be considered for the search for biomarkers for the prevention of suicide behavior in future studies.

INTRODUCTION

1- Other biological factors associated with suicide behavior should be presented in the introduction. I recommend to expand the first part concerning the association between hormones, methylation and suicide.

Response:

Changes in the manuscript

Page 2, line 55-59

The literature has found some biological factors associated with suicidal behavior, such as low function of the serotonergic systems that were presented in people with suicidal acts (Mann 2013). As well as dopamine system is associated with suicide attempt risk in some individuals (Hoertel, et al. 2021; Institute of Medicine Committee on, et al. 2002). Also, studies demonstrate that chronic stress has been associated with alterations in the HPA axis and suicide behavior [4-7] .

2-The main objective of the paper should be better clarified. I recommend to add a subsection called 1.1. Aims to clarify them.

Response: Thank you so much for your recommendations, we added that subsection.

Changes in the manuscript

Page 2, line 85-88

  • Aims to clarify

Additionally, we also aimed to determine if the DNA methylation levels of genes that participates HPA axis activation could be associated with an increased risk of suicide behavior.

MATERIAL AND METHODS

1- In section 2.2 the authors report the inclusion criteria and variables recorded. What about exclusion criteria?

Response: Thank you so much, we added exclusion criterio

Changes in the manuscript

Page 3, line 117-120

2.3. Exclusion criteria

The exclusion criteria were: duplicated publications, studies of cases only or case reports, and papers that did not have enough data available.

RESULTS

1- Section 3.3. should not be presented as a separate section with figures and tables. They should be presented in the body of the manuscript where appropiate. 

Response: Thank you so much for your observations, we changed the figures and tables in the body of the manuscript.

2- PRISMA flowchart should be presented in the methods section. This figure should be earlier presented.

Response: Thank you so much for all your observations, we changed the flowchart in the methods section.

CONCLUSIONS

1- A separate section for the conclusions is needed. Conclusions should not be referencing a Figure.

Response: Thank you so much for your comment. We added a separate section.

Reviewer 3 Report

My suggestions:

1. Are the suicide-related genes, the authors mentioned share some common pathways? The authors may draw a figure in the discussion, which shows their common mechanisms (if there is).

2. Was any article mentioned, whether the children had OCD? 

3. Tables may be needed to be better organized since they are a little confusing. 

4. In the introduction, the authors may discuss depression and the risk factors for depression in a little bit more in detail.

5.  Were there studies on histone acetylation or miRNAs, which could impact depression risk or treatment? You may mention it briefly. 

Author Response

REVIEWER 3

My suggestions:

  1. Are the suicide-related genes, the authors mentioned share some common pathways? The authors may draw a figure in the discussion, which shows their common mechanisms (if there is).

Response: Thank you so much for your suggestion.

Changes in the manuscript

Page 10, line 347-349

Due to genes that have changes in their methylation levels are involved in the activation of the HPA axis and in the creation of the hormone cortisol, they could be share this common pathway.

  1. Was any article mentioned, whether the children had OCD? 

Response: Thank you so much for your question, we searched and didn’t find this information.

  1. Tables may be needed to be better organized since they are a little confusing. 

Response: Thank for your suggestion, we reorganized all the tables.

  1. In the introduction, the authors may discuss depression and the risk factors for depression in a little bit more in detail.

Response: Thank you so much for your suggestion, we added information about risk factors.

Change in the manuscript

Page 2, line 85-88

Major depressive disorder is an important disabling mental disorder, and some risk factors that are associated with the development of MDD[4] are a family history of MDD [5] , deficits in how individuals process reward [6], substance use disorder [7], sexual abuse [8], social isolation and others [9]

  1. Were there studies on histone acetylation or miRNAs, which could impact depression risk or treatment? You may mention it briefly. 

Response: Thank you so much for your comment, we mentioned the studies.

Change in the manuscript

Page 2, line 90-97

In the case of depression, some epigenetic mechanisms could mediate the risk to have depression, not only DNA methylation otherwise microRNAs (miRNAs) and histone modifications. (Penner-Goeke and Binder 2019; Uchida, et al. 2018). For example, studies showed altered levels of specific miRNA in the peripheral tissue of patients with MDD(Yuan, et al. 2018). And in brain tissue another study found miR-1202 downregulated in the PFC of MDD patients who died by suicide (Lopez, et al. 2014). As for histone modifications, found increased global acetylation of histone 3 at lysine 14 (H3K14ac) in the nucleus accumbens of patients with MDD and reported downregulation of HDAC2 (Covington, et al. 2009).

Round 2

Reviewer 1 Report

All comments from the Reviewer have been included in the current version of the manuscript

Reviewer 3 Report

Authors fulfilled my suggestions, thank you.